# CADDA: Class-wise Automatic Differentiable Data Augmentation for EEG Signals

**Cédric Rommel, Thomas Moreau, Joseph Paillard & Alexandre Gramfort**
Université Paris-Saclay, Inria, CEA, Palaiseau, 91120, France
`{firstname.lastname}@inria.fr`

## Abstract

Data augmentation is a key element of deep learning pipelines, as it informs the network during training about transformations of the input data that keep the label unchanged. Manually finding adequate augmentation methods and parameters for a given pipeline is however rapidly cumbersome. In particular, while intuition can guide this decision for images, the design and choice of augmentation policies remains unclear for more complex types of data, such as neuroscience signals. Besides, class-dependent augmentation strategies have been surprisingly unexplored in the literature, although it is quite intuitive: changing the color of a car image does not change the object class to be predicted, but doing the same to the picture of an orange does. This paper investigates gradient-based automatic data augmentation algorithms amenable to class-wise policies with exponentially larger search spaces. Motivated by supervised learning applications using EEG signals for which good augmentation policies are mostly unknown, we propose a new differentiable relaxation of the problem. In the class-agnostic setting, results show that our new relaxation leads to optimal performance with faster training than competing gradient-based methods, while also outperforming gradient-free methods in the class-wise setting. This work proposes also novel differentiable augmentation operations relevant for sleep stage classification.

## 1 Introduction

The interest in using deep learning for EEG related tasks has been rapidly growing in the last years, specially for applications in sleep staging, seizure detection and prediction, and brain-computer interfaces (BCI) (Roy et al., 2019). Data augmentation is a well-known regularization technique, widely used to improve the generalization power of large models, specially in deep learning (Krizhevsky et al., 2012; Yaeger et al., 1996; Simard et al., 2003). Not only does it help by synthetically increasing the size of the dataset used for training, it also creates useful inductive biases, as it encodes invariances of the data and the underlying decision function which the model does not have to learn from scratch (Chen et al., 2020a; van der Maaten et al., 2013). Such invariant transforms are also a key ingredient for state-of-the-art self-supervised learning (Chen et al., 2020b). Unfortunately, these transforms have to be known *a priori* and the best augmentations to use often highly depend on the model architecture, the task, the dataset and even the training stage (Ho et al., 2019; Cubuk et al., 2020). Manually finding what augmentation to use for a new problem is a cumbersome task, and this motivated the proposal of several automatic data augmentation search algorithms (Cubuk et al., 2019).

The existing automatic data augmentation literature often focuses on computer vision problems only, and its application to other scientific domains such as neuroscience has been under-explored. Data augmentation is all the more important in this field, as brain data, be it functional MRI (fMRI) or electroencephalography (EEG) signals, is very scarce either because its acquisition is complicated and costly or because expert knowledge is required for labelling it, or both. Furthermore, while atomic transformations encoding suitable invariances for images are intuitive (if you flip the picture of a cat horizontally it is still a cat), the same cannot be said about functional brain signals such as EEG. Hence, automatic data

augmentation search could be helpful not only to improve the performance of predictive models on EEG data, but also to discover interesting invariances present in brain signals.

Another interesting aspect of data augmentation that has gotten little attention is the fact that suitable invariances often depend on the class considered. When doing object recognition on images, using color transformations during training can help the model to better recognize cars or lamps, which are invariant to it, but will probably hurt the performance for classes which are strongly defined by their color, such as apples or oranges. This also applies to neuroscience tasks, such as sleep staging which is part of a clinical exam conducted to characterize sleep disorders. As most commonly done (Iber et al., 2007), it consists in assigning to windows of 30 s of signals a label among five: Wake (W), Rapid Eye Movement (REM) and Non-REM of depth 1, 2 or 3 (N1, N2, N3). While some sleep stages are strongly characterized by the presence of waves with a particular shape, such as spindles and K-complexes in the N2 stage, others are defined by the dominating frequencies in the signal, such as alpha and theta rhythms in W and N1 stages respectively (Rosenberg & Van Hout, 2013). This means that while randomly setting some small portion of a signal to zero might work to augment W or N1 signals, it might wash out important waves in N2 stages and slow down the learning for this class. This motivates the study of augmentations depending on the class. Of course, as this greatly increases the number of operations and parameters to set, handcrafting such augmentations is not conceivable and efficient automatic searching strategies are required, which is the central topic of this paper. Using black-box optimization algorithms as most automatic data augmentation papers suggest seemed unsuitable given the exponential increase in complexity of the problem when separate augmentations for each class are considered.

In this paper, we extend the bilevel framework of AutoAugment (Cubuk et al., 2019) in order to search for class-wise (CW) data augmentation policies. First, Section 3 introduces three novel augmentation operations for EEG signals, and Section 4 quantifies on sleep staging and digit classification tasks how CW augmentations can enable gains in prediction performance by exploiting interesting invariances. Then, Section 5 introduces a novel differentiable relaxation of this extended problem which enables gradient-based policy learning. Finally, in Section 6, we use the EEG sleep staging task in the class-agnostic setting to evaluate our approach against previously proposed gradient-based methods. In the class-wise setting, the CADDA method is compared against gradient-free methods that can suffer significantly from the dimension of policy learning problem. Furthermore, we carry an ablation study which clarifies the impact of each architecture choices that we propose. Our experiments also investigate density matching-based approaches (Lim et al., 2019; Hataya et al., 2020) in low or medium data regimes.

## 2 Related Work

**EEG Data Augmentation** Given the relatively small size of available EEG datasets, part of the community has explored ways of generating more data from existing ones, *e.g.,* using generative models (Hartmann et al., 2018; Bouallegue & Djemal, 2020) or data augmentation strategies (*e.g.,* Roy et al. 2019; Yin & Zhang 2017; Wang et al. 2018). Here, we give a succinct review which is completed in Appendix B. The reader is referred to Roy et al. (2019) for a more detailed discussion on previous EEG data augmentation papers.

Noise addition is the most straight-forward data augmentation that can be applied to either raw EEG signals (Wang et al., 2018) or to derived features (Yin & Zhang, 2017). Adding such transformed samples forces the estimator to learn a decision function that is invariant to the added noise. Other transforms have also been proposed to account for other sources of noise, such as label misalignment with the `time shift` (Mohsenvand et al., 2020), positional noise for the sensors with `sensor rotations` (Krell & Kim, 2017) or corrupted sensors with `channel dropout` (Saeed et al., 2021).

Other data augmentations aim at promoting some global properties in the model. While masking strategies such as `time masking`, `bandstop filter` (Mohsenvand et al., 2020) or `sensors cutout` (Cheng et al., 2020) ensure that the model does not rely on specific time segments, frequency bands or sensor, `channel symmetry` (Deiss et al., 2018) encourages the

model to account for the brain bilateral symmetry. Likewise, the `Fourier Transform (FT) surrogate` (Schwabedal et al., 2019) consists in replacing the phases of Fourier coefficients by random numbers sampled uniformly from $[0, 2\pi)$. The authors of this transformation argue that EEG signals can be approximated by linear stationary processes, which are uniquely characterized by their Fourier amplitudes.

**Automatic Data Augmentation** Automatic data augmentation (ADA) is about searching augmentations that, when applied during the model training, will minimize its validation loss, leading to greater generalization. Let $D_{\text{train}}$ and $D_{\text{valid}}$ denote a training and validation set respectively, and let $\mathcal{T}$ be an augmentation policy, as defined in more detail in Section 4. ADA is about finding algorithms solving the following bilevel optimization problem:

$$
\begin{aligned}
\min_{\mathcal{T}} \quad & \mathcal{L}(\theta^* | D_{\text{valid}}) \\
\text{s.t.} \quad & \theta^* \in \arg\min_{\theta} \mathcal{L}(\theta | \mathcal{T}(D_{\text{train}})),
\end{aligned}
\tag{1}
$$

where $\theta$ denotes the parameters of some predictive model, and $\mathcal{L}(\theta | D)$ its loss over set $D$.

One of the first influential works in this area is AutoAugment (Cubuk et al., 2019), where problem (1) is solved by fully training multiple times a smaller model on a subset of the training set with different augmentation policies and using the validation loss as a reward function in a reinforcement learning setting. The main drawback of the method is its enormous computation cost. Many alternative methods have been proposed since, differing mainly in terms of search space, search algorithm, and metric used to assess each policy.

The first attempts to make AutoAugment more efficient consisted in carrying model and policy trainings jointly, as done with a genetic algorithm in Population-Based Augmentation (Ho et al., 2019). A different way of alleviating the computation burden of AutoAugment is proposed in Tian et al. (2020). Observing that data augmentation is mostly useful at the end of training, the authors propose to pre-train a shared model close to convergence with augmentations sampled uniformly, and then to use it to warmstart AutoAugment.

The previous methods (Cubuk et al., 2019; Ho et al., 2019; Lim et al., 2019) use proxy tasks with small models and training subsets to carry the search. This idea is challenged in RandAugment (Cubuk et al., 2020), where it is shown that optimal augmentations highly depend on the dataset size and model. RandAugment simply samples augmentations uniformly with the same shared magnitude, which can be tuned with a grid-search. Competitive results are obtained on computer vision tasks with this naive policy. A similar approach is proposed in Fons et al. (2021), where all possible augmentations are weighted with learnable parameters and used to derive enlarged batches.

**Density Matching** While all previously cited ADA methods try to solve in some sense the original problem (1), Fast AutoAugment (Lim et al., 2019) suggests to solve a surrogate problem, by moving the policy $\mathcal{T}$ into the upper-level:

$$
\begin{aligned}
\min_{\mathcal{T}} \quad & \mathcal{L}(\theta^* | \mathcal{T}(D_{\text{valid}})) \\
\text{s.t.} \quad & \theta^* \in \arg\min_{\theta} \mathcal{L}(\theta | D_{\text{train}}) \ .
\end{aligned}
\tag{2}
$$

Problem (2) can be seen as a form of *density matching* (Lim et al., 2019), where we look for augmentation policies such that the augmented validation set has the same distribution as the training set, as evaluated through the lens of the trained model. This greatly simplifies problem (1) which is no longer bilevel, allowing to train the model only once without augmentation. Computation is massively reduced, yet this simplification assumes the trained model has already captured meaningful invariances. This density matching objective has been later reused in Faster AutoAugment (Hataya et al., 2020), where a Wasserstein GAN network (Arjovsky et al., 2017) is used instead of the trained classifier to assess the closeness between augmented and original data distributions.

**Gradient-based Automatic Data Augmentation** Further efficiency improvements in ADA were obtained by exploring gradient-based optimization. In Online Hyper-parameter Learning (Lin et al., 2019) and Adversarial AutoAugment (Zhang et al., 2020), policies are

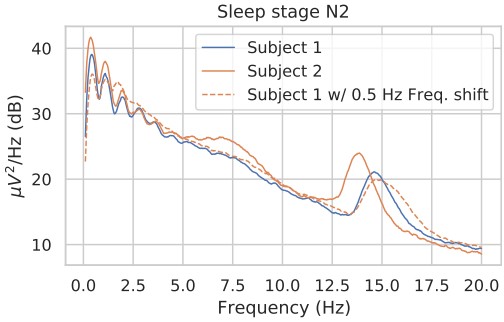

Figure 1: Averaged power spectral density of N2 windows from one night sleep of two different subjects from the sleep Physionet dataset (channel Pz-Oz used here) (Goldberger et al., 2000). We notice that peak frequencies are shifted. Applying a 0.5 Hz `frequency shift` transform to subject 1 leads a power spectrum density more similar to subject 2.

modeled as parametrized discrete distributions over possible transformations and updated using the REINFORCE gradient estimator (Williams, 1992). As such estimators are quite noisy, Faster AutoAugment (*Faster AA*) and DADA (Li et al., 2020) derive full continuous relaxations of the discrete original formalism of AutoAugment, allowing them to backpropagate directly through policies. We have revisited this idea in this work.

**Class-dependent Data Augmentation** While class-dependent data generation has been studied in the GAN literature (Mirza & Osindero, 2014), to our knowledge, Hauberg et al. (2016) is the only work that has explored class-dependent data augmentations. It presents a method for learning a distribution of CW spatial distortions, by training a model to find the $C^1$-diffeomorphism allowing to transform one example into another within the same class. Although the authors state it is applicable to other domains, it is only demonstrated for digit classification and its extension to other frameworks seems non-trivial. The main difference with our work is that Hauberg et al. (2016) learns transformations from scratch, while we try to learn which one to pick from a pool of existing operations and how to aggregate them.

## 3 NEW EEG DATA AUGMENTATIONS

In addition to the data augmentations described in Section 2, we investigate in this paper three novel operations acting on the time, space and frequency domains.

**Time Reverse** As a new time domain transformation, we propose to randomly reverse time in certain input examples (*i.e.,* flip the time axis of the signal). Our motivation is the belief that most of the sleep stage information encoded in EEG signals resides in relative proportions of frequencies and in the presence of certain prototypical short waves. Yet, on this last point, it is possible that not all sleep stages are invariant to this transform, as some important waves (*e.g.,* K-complexes) are asymmetric and are therefore potentially altered by the transformation.

**Sign Flip** In the spatial transformations category, we argue that the information encoded in the electrical potentials captured by EEG sensors are likely to be invariant to the polarity of the electric field. Given the physics of EEG, the polarity of the field is defined by the direction of the flow of electric charges along the neuron dendrites: are charges moving towards deeper layers of the cortex or towards superficial ones? As both are likely to happen in a brain region, we propose to augment EEG data by randomly flipping the signals sign (multiplying all channels' outputs by −1). This can be interpreted as a spatial transformation, since in some configurations it corresponds to inverting the main and the reference electrode (e.g. in the Sleep Physionet dataset).

**Frequency Shift** Reading the sleep scoring manual (Rosenberg & Van Hout, 2013), it is clear that dominant frequencies in EEG signals play an important role in characterising different sleep stages. For instance, it is said that the N2 stage can be recognized by the presence of sleep spindles between 11-16 Hz. Such frequency range is quite broad, and when looking at the averaged power spectral density of windows in this stage for different individuals, one can notice that the frequency peaks can be slightly shifted (*cf.* Figure 1). To mimic this phenomena we propose to shift the frequencies of signals by an offset $\Delta f$ sampled uniformly from a small range (See Appendix B for details).

## 4    CLASS-WISE DATA AUGMENTATION

**Background on auto augmentation framework** We adopt the same framework of definitions from AutoAugment (Cubuk et al., 2019), which was also reused in Lim et al. (2019) and Hataya et al. (2020). Let $\mathcal{X}$ denote the inputs space. We define an *augmentation operation* $\mathcal{O}$ as a mapping from $\mathcal{X}$ to $\mathcal{X}$, depending on a magnitude parameter $\mu$ and a probability parameter $p$. While $\mu$ specifies how strongly the inputs should be transformed, $p$ sets the probability of actually transforming them:

$$\mathcal{O}(X; p, \mu) := \begin{cases} \mathcal{O}(X; \mu) & \text{with probability } p \quad , \\ X & \text{with probability } 1 - p \quad . \end{cases} \tag{3}$$

Operations can be chained, forming what we call an *augmentation subpolicy* of length $K$:

$$\tau(X; \mu_\tau, p_\tau) := (\mathcal{O}_K \circ \cdots \circ \mathcal{O}_1)(X; \mu_\tau, p_\tau),$$

where $\mu_\tau$ and $p_\tau$ denote the concatenation of the $K$ operations parameters. To increase the randomness of the augmentation strategy, $L$ subpolicies are grouped into a set called an *augmentation policy* $\mathcal{T}$, and sampled with uniform probability for each new batch of inputs:

$$\mathcal{T}(X) = \tau_i(X; \mu_{\tau_i}, p_{\tau_i}), \quad \text{with } i \sim \mathcal{U}(\{1, \ldots, L\}). \tag{4}$$

Note that some operations, such as `time reverse` and `sign flip` augmentations described in Section 3, may not depend on a magnitude parameter, which reduces the search space.

**Novel class-wise subpolicies and policies** We introduce in this paper augmentation subpolicies which are conditioned on the input class. Hence, we define a *class-wise subpolicy* as a mapping between inputs $X$ and labels $y$ to an augmentation subpolicy $\tau_y$:

$$\tilde{\tau} : (X, y) \mapsto \tau_y(X) \tag{5}$$

where $\{\tau_y : y \in \mathcal{Y}\}$ is a set of subpolicies for each possible class in the output space $\mathcal{Y}$. These can be grouped into *class-wise policies* as in (4).

**Selection for class-wise augmentations** In order to illustrate the interest of CW augmentations, we explore their performances on a sleep staging task using the sleep Physionet dataset (Goldberger et al., 2000). Following the standardization and low-pass filtering of the the data, 30-seconds labelled time windows were extracted from the recordings. To avoid observing the effects due to class imbalance, we sub-sampled the windows to obtain a balanced dataset. Indeed, as data augmentation yields more significant enhancements in low data regime, it tends to benefit underrepresented classes more than others. Out of 83 subjects, 8 were left out for testing and the remaining ones were then split in training and validation sets, with respective proportions of 0.8 and 0.2. The convolutional neural network from Chambon et al. (2018) [1] was then trained on 350 randomly selected windows from the training set using CW subpolicies made up of two different augmentations  and this was repeated 10 times using different folds. We carried this out for all possible combinations of two augmentations and kept  the best CW subpolicy based on the cross- validation score compared to the class-agnostic augmentations. The aforementioned model was finally evaluated on the test set.

Figure 2 (a) reports the improvement of the per-class F1 score for `channels dropout`, `sign flip` and a CW subpolicy which augments W and N3 stages with `channels dropout` and N1, N2 and REM with `sign flip`. The relative improvement in overall balanced accuracy compared to only augmenting with the class-agnostic augmentations is depicted on Figure 2 (b), which shows that a better performance is reached with the CW augmentation.

This experiment suggests that looking for augmentation strategies depending on the label of the data being transformed can be relevant to improve prediction performance, but also to help discovering some interesting invariances for the task considered. A similar toy experiment with MNIST dataset (LeCun et al., 1998) can be found in Appendix C with automated CW-policy selection using Random Search.

---

[1]See Appendix D for a detailed description of the architecture.

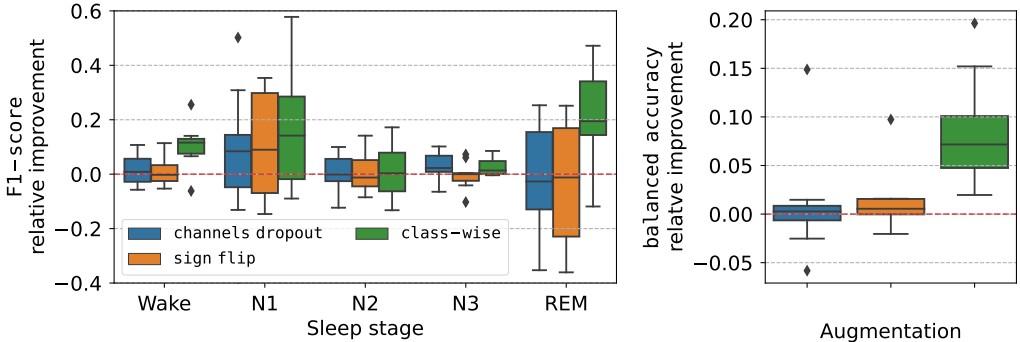

Figure 2: (a) Improvement per class of the F1-score due to `channels dropout`, `sign flip` and the CW subpolicy, relative to a model trained with no data augmentation. Each augmentation encodes specific invariances that can be more relevant for some classes than others. (b) Improvement of the multi-class balanced-accuracy relative to a model trained with no data augmentation. The CW subpolicy outperforms class-agnostic ones. Both boxplots show how values are spread out across 10 folds.

## 5 EFFICIENT AUTOMATIC DATA AUGMENTATION USING GRADIENTS

Although the previous experiment in Figure 2 motivates the interest in CW data augmentation, it also illustrates how impractical it can be to search such augmentations manually for each new task, dataset and model. Moreover, considering $N_\mu$ possible magnitudes, $N_\mathcal{O}$ operations, $N_p$ probability values and $|\mathcal{Y}|$ ($|\mathcal{Y}| = 1$ in the class-agnostic case) possible classes leads to $(N_p \times N_\mu \times N_\mathcal{O})^{L \times K \times |\mathcal{Y}|}$ possible policies, which becomes very rapidly a huge space to explore, especially with gradient-free algorithms. This is typically illustrated in Appendix C where even though the setting is quite simple, Random Search requires a very large number of draws to find interesting policies. Therefore, it is unlikely that such algorithms can scale well when considering CW policies, which considerably increase the search space size. This motivates the exploration of gradient-based search approaches, which can select policies more efficiently in such a huge search space.

### 5.1 DIFFERENTIABLE AUGMENTATION POLICIES

Most ADA approaches are based on gradient-free algorithms, mainly because of the discrete structure of augmentation policies (Section 4). For this reason, Faster AA (Hataya et al., 2020) and DADA (Li et al., 2020) propose continuous relaxations of the AutoAugment policies, inspired by a recent Neural Architecture Search (NAS) method: DARTS (Liu et al., 2019). Hereafter we build on these relaxation ideas in the EEG setting.

**Background on probabilities and magnitudes relaxation** Concerning probabilities, equation (3) is relaxed using a Relaxed Bernoulli random variable (Maddison et al., 2017) as in Hataya et al. (2020) and Li et al. (2020) (*cf.* Appendix A for further details). As for magnitudes, while some operations don't depend on any magnitude parameter, most of the augmentations described in section Section 2 do and are not differentiable with respect to it. It is argued in Hataya et al. (2020) that using a simple *straight-through gradient estimator* (Bengio et al., 2013) is enough to face this difficulty (*i.e.,* approximating the gradient of $\mathcal{O}(X; \mu)$ w.r.t $\mu$ by 1). In practice, we found that this approach alone did not allow to properly transmit the gradient loss information to tune the magnitudes. For this reason, we preferred to carry out a case-by-case relaxation of each operations considered, which is detailed in Appendix B.

**Novel operations relaxation** Operations composing subpolicies also need to be selected over a discrete set. For this, Faster AA replaces the sequence of operations by a sequence of

stages $\{\tilde{\mathcal{O}}_k : k = 1, \ldots, K\}$, which are a convex combination of all operations:

$$\tilde{\mathcal{O}}_k(X) = \sum_{n=1}^{N_{\mathcal{O}}} [\sigma_\eta(w_k)]_n \mathcal{O}_k^{(n)}(X; \mu_k^{(n)}, p_k^{(n)}), \tag{6}$$

where $w_k$ denote the weights vector of stage $k$. These weights are passed through a softmax activation $\sigma_\eta$, so that, when parameter $\eta$ is small, $\sigma_\eta(w_k)$ becomes a onehot-like vector.

We used the same type of architecture, but replaced the softmax $\sigma_\eta$ by a straight-through Gumbel-softmax distribution (Jang et al., 2017) parametrized by $\{w_k\}$. As shown in experiments of Section 6, this allows to gain in efficiency, as we only sample one operation at each forward pass and do not need to evaluate all operations each time. Furthermore, given that the original Gumbel-softmax gradients are biased, we use the unbiased RE-LAX gradient estimator (Grathwohl et al., 2018). The same idea is used in DADA (Li et al., 2020), except that they consider a different policy architecture where they sample whole subpolicies instead of operations within subpolicies. These architecture choices are compared in our ablation study in Section 6 and Appendix E.

**Algorithm 1:** (C)ADDA

**Input :** $\xi, \epsilon > 0$, Datasets $D_{\text{train}}, D_{\text{valid}}$,
            Trainable policy $\mathcal{T}_\alpha$, Model $\theta$
**Result:** Policy parameters $\alpha$
**while** *not converged* **do**
  // compute the unrolled model
  $g_\theta = \mathcal{L}(\theta|\mathcal{T}_\alpha(D_{\text{train}})).\text{backward}(\theta)$
  $\theta' := \theta - \xi g_\theta$
  // Estimate $\nabla_\alpha \mathcal{L}(\theta'|D_{\text{valid}})$
  $g'_\theta = \mathcal{L}(\theta'|D_{\text{valid}}).\text{backward}(\theta)$
  $g_\alpha^+ = \mathcal{L}(\theta + \epsilon g'_\theta|\mathcal{T}_\alpha(D_{\text{train}})).\text{backward}(\alpha)$
  $g_\alpha^- = \mathcal{L}(\theta - \epsilon g'_\theta|\mathcal{T}_\alpha(D_{\text{train}})).\text{backward}(\alpha)$
  $g_\alpha = \frac{1}{2\epsilon}(g_\alpha^+ - g_\alpha^-)$
  // Update Policy parameters $\alpha$
  $\alpha = \alpha - \xi g_\alpha$
  // Update the model parameters $\theta$
  $g_\theta = \mathcal{L}(\theta|\mathcal{T}_\alpha(D_{\text{train}})).\text{backward}(\theta)$
  $\theta = \theta - \xi g_\theta$
**end**

### 5.2 Policy optimization

While Faster AA casts a surrogate density matching problem (2), which is easier and faster to solve then (1), Tian et al. (2020) presents empirical evidence suggesting that these metrics are not really correlated. Hence, we propose to tackle directly the bilevel optimization from (1) as done in DADA and DARTS.

Let $\alpha$ be the vector grouping all the parameters of a continuously relaxed policy model $\mathcal{T}_\alpha$ (Section 5.1). We carry alternating optimization steps of the policy parameters $\alpha$ and the model parameters $\theta$ as described in Algorithm 1. In order to estimate the augmentation policy gradient, we approximate the lower-level of (1) by a single optimization step:

$$\nabla_\alpha \mathcal{L}(\theta^*|D_{\text{valid}}) \simeq \nabla_\alpha \mathcal{L}(\theta'|D_{\text{valid}}), \qquad \text{with} \quad \theta' := \theta - \xi \nabla_\theta \mathcal{L}(\theta|\mathcal{T}_\alpha(D_{\text{train}})), \tag{7}$$

where $\xi$ denotes the learning rate. By applying the chain rule in Equation (7) we get:

$$\nabla_\alpha \mathcal{L}(\theta'|D_{\text{valid}}) = -\xi \nabla_{\alpha,\theta}^2 \mathcal{L}(\theta|\mathcal{T}_\alpha(D_{\text{train}})) \nabla_{\theta'} \mathcal{L}(\theta'|D_{\text{valid}}) \tag{8}$$

$$\simeq -\xi \frac{\nabla_\alpha \mathcal{L}(\theta^+|\mathcal{T}_\alpha(D_{\text{train}})) - \nabla_\alpha \mathcal{L}(\theta^-|\mathcal{T}_\alpha(D_{\text{train}}))}{2\epsilon}, \tag{9}$$

where $\epsilon$ is a small scalar and $\theta^\pm = \theta \pm \epsilon \nabla_\theta \mathcal{L}(\theta'|D_{\text{valid}})$. The second line (9) corresponds to a finite difference approximation of the Hessian-gradient product.

## 6 Experiments on EEG data during sleep

**Datasets** We used the public dataset MASS - Session 3 (O'reilly et al., 2014). It corresponds to 62 nights, each one coming from a different subject. Out of the 20 available EEG channels, referenced with respect to the A2 electrode, we used 6 (C3, C4, F3, F4, O1, O2). We also used the standard sleep Physionet data (Goldberger et al., 2000) which contains 153 recordings from 83 subjects. Here two EEG derivations are available (FPz-Cz and Pz-Oz). For both datasets, sleep stages were annotated according to the AASM rules (Iber et al., 2007). The EEG time series were first lowpass filtered at 30 Hz (with 7 Hz transition bandwidth) and

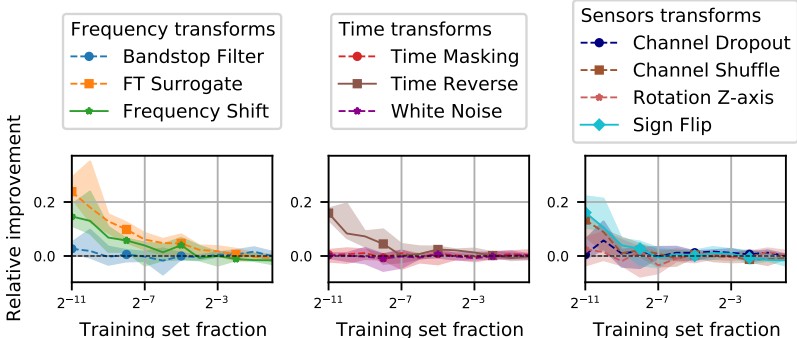

Figure 3: Median performance gains obtained by individual augmentation strategies on the Physionet dataset while increasing the train set size. Results are relative improvements compared to the baseline without any augmentation. Two frequency transforms (`FT surrogate` and `frequency shift`) as well as `time reverse` lead to the best performance.

standardized. For MASS they were resampled from 256 Hz to 128 Hz, while the Physionet dataset was kept at 100 Hz. Further experimental details can be found in Appendix D.

**Manual exploration** All EEG data augmentation techniques were first tested individually on the two datasets without an automatic search strategy. The Physionet dataset was split in 5 folds with 16 subjects for Physionet and 12 subjects for MASS in a test set. Among the training folds, data were split in 80/20 randomly to obtain a training and a validation set. To assess the effect of augmentation at different data regimes, the training set was sequentially subset using a log2 scale (with stratified splits). All augmentations tested had 0.5 probability and a manually fine-tuned magnitude (*cf.* Section G.2). Results on Physionet are presented on Figure 3, while similar plots for MASS can be found in the appendix (Figure G.8). As expected, augmentation plays a major role in the low data regime, where it teaches invariances to the model which struggles to learn them from the small amount of data alone. Transforms like `FT surrogate`, `frequency shift` and `time reverse` lead to up to 20% performance improvement on the test set for Physionet and 60% for MASS in extremely low data regimes. Surprisingly, `rotations` and `time masking` do not seem to help a lot.

**More efficient gradient-based search** As commonly done (Cubuk et al., 2019; Ho et al., 2019; Lim et al., 2019; Hataya et al., 2020), we considered policies of size $L = 5$, made of subpolicies of length $K = 2$ containing $N_\mathcal{O} = 12$ operations (*cf.* Appendix D). The MASS dataset was used for this experiment. Both the training and validation sets consisted of 24 nights each, and the test set contained 12 nights. For each method, we stopped the search every given number of steps (2 epochs or 5 samplings), used the learned policy to retrain from scratch the model (leading to a point in Figure 4) and resumed the search from where it stopped. For each run, when the final retraining validation accuracy improves compared to previous retraining, we report the new test accuracy obtained by the retrained model (otherwise, we keep the previous value). Figure 4 (a) shows that our novel *automatic differentiable data augmentation* method (ADDA) outperforms existing gradient-based approaches both in speed and final accuracy. In Figure F.4, a comparison with other gradient-free approaches in a class-agnostic setting is also provided, where ADDA is shown to outperform the state-of-the-art in speed and accuracy.

**Ablation study of differentiable architecture in the class-agnostic setting** In order to better understand previous results, we carried out an ablation study in the class-agnostic setting, where we sequentially removed from ADDA (green): the RELAX gradient estimator (red), followed by the Gumbel-softmax sampling (light blue). We see in Figure 4 (a) that RELAX is mainly responsible for the final accuracy increase, as it removes the gradients' bias induced by the Gumbel-softmax sampling. We also clearly see from curves in red and light blue that replacing the softmax $\sigma_\eta$ from the subpolicy stages (6) by a Gumbel-softmax sampling is responsible for the considerable 4x speed-up. Note that the core difference

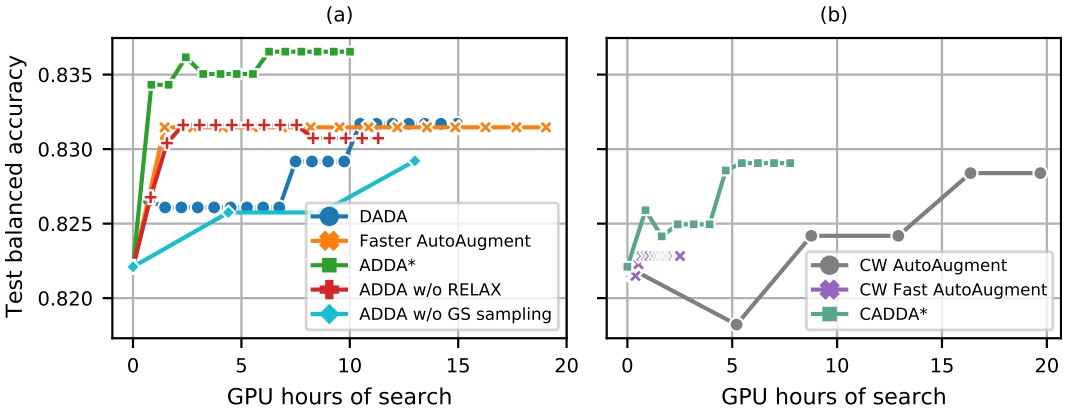

Figure 4: Median performance (over 5 folds) of different ADA strategies as a function of the computation time. (a) Class-agnostic setting: ADDA is 40% faster than Faster AA and reaches a performance 0.6% higher. It also outperforms DADA by 0.6% in accuracy, 4 GPU hours earlier. (b) CW setting: CADDA outperforms gradient-free methods in this setting, as it is 5x faster than AutoAugment and achieves higher performance than Fast AutoAugment.

between ADDA and DADA is the fact that the former samples operations within each subpolicy, and the latter samples whole subpolicies. Curves in green and dark blue show that ADDA converges faster, which means the first choice is more efficient. This might be due to the fact that our architecture expresses the same possibilities with $L \times K \times N_{\mathcal{O}} = 120$ weights $w_k$, which in our case is less than the $(N_{\mathcal{O}})^K = 144$ weights in DADA. Most importantly, note that during the training DADA will sample and update the probabilities and magnitudes of subpolicies with higher weights more often, while ADDA keeps updating all its subpolicies with equal probability. This allows ADDA to converge faster to 5 good subpolicies while DADA can get stuck with only one or two strong ones.

**Efficient search in class-wise setting** Figure 4 (b) compares the best performing gradient-based approach in the class-wise setting, namely CADDA, against state-of-the-art gradient-free methods (AutoAugment and Fast AutoAugment) optimized with TPE (Bergstra et al., 2011), as implemented in optuna (Akiba et al., 2019). CADDA achieves top performance while being significantly faster than AutoAugment. However, one can observe on this CW setting that CADDA does not improve over ADDA in the class-agnostic case. This illustrates the difficulty of learning CW policies due to the dimensionality of the search space. We also hypothesize that this could reflect the difficulty of the simple CNN model considered here to encode the complex invariances promoted by the CW augmentations.

CONCLUDING REMARKS

Our work explores the problem of automatic data augmentation beyond images which is still rarely considered in the literature. We provided novel transforms for EEG data, and a state-of-the-art search algorithm – called (C)ADDA – to address this question. This method yields very promising results for sleep stage classification, with up to 40% speed up and superior accuracy compared to existing gradient-based methods in a standard setting. Motivated by the possible impact of CW policies that we illustrated empirically, we proposed a framework for the automatic search of such augmentations. Our work shows that gradient-based automatic augmentation approaches are possible and necessary in the CW setting due to the size of the search space. Finally, while CW does not provide the expected performance boost yet, we believe that this framework opens a novel avenue to improve the field of data augmentation, in particular for neuroscience applications.

ACKNOWLEDGMENTS AND DISCLOSURE OF FUNDING

This work was supported by the BrAIN grant (ANR-20-CHIA-0016). It was also granted access to the HPC resources of IDRIS under the allocation 2021-AD011012284 and 2021-AD011011172R1 made by GENCI.

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

# A  DETAILS ON OPERATIONS RELAXATION

Hereafter are some general explanations concerning the continuous relaxation of augmentation operations.

## A.1  RELAXATION REGARDING PROBABILITIES

Concerning the probabilities, equation (3) can be rewritten as follows:

$$\mathcal{O}(X; p, \mu) = b\mathcal{O}(X; \mu) + (1 - b)X \ ,  \tag{10}$$

where $b$ is sampled from a Bernoulli distribution with probability $p$. As the Bernoulli distribution is discrete, equation (10) can be made differentiable with respect to $p$ by using a Relaxed Bernoulli random variable (Maddison et al., 2017), *i.e.,* by reparametrizing the distribution as a continuous function depending on $p$ and on a uniform random variable (Schulman et al., 2015).

## A.2  RELAXATION REGARDING MAGNITUDES

As for magnitudes, while some operations don't depend on any magnitude parameter, most of the augmentations described in section Section 2 do and are not differentiable with respect to it. It is argued in Hataya et al. (2020) that using a simple *straight-through gradient estimator* (Bengio et al., 2013) is enough to face this difficulty (*i.e.,* approximating the gradient of $\mathcal{O}(X; \mu)$ w.r.t $\mu$ by 1). In practice, we found that this approach alone did not allow to properly transmit the gradient loss information to tune the magnitudes (*cf.* Section A.3). For this reason, we preferred to carry out a case-by-case relaxation of each operations considered (*cf.* Appendix B). Overall, the same relaxation principles were used in most of them:

1. when $\mu$ is used to set some sampling operation, a pathwise gradient estimator is used (Schulman et al., 2015), just as for $p$ in Eq. (10);

2. when some masking is necessary, vector indexing is replaced by element-wise multiplications between the transformed vector and a smooth mask built using sigmoid functions;

3. the straight-through estimator is only used to propagate the gradient through the permutation operation in `channel shuffle`.

## A.3  FURTHER EXPLANATION CONCERNING STRAIGHT THROUGH MAGNITUDE GRADIENT ESTIMATOR

In order to validate the continuous relaxation of each augmentation operation described in Appendix B, we tried to *fit* the identity, *i.e.,* we minimized the mean squared error between augmented and unchanged batches of EEG signals:

$$\min_{p, \mu} \|X - \mathcal{O}(X; p, \mu)\|^2.$$

This allowed us to realize that using a simple straight-through estimator as suggested by Hataya et al. (2020) was not enough to learn good magnitude parameters. Indeed, we observed with many different learning rates that $\mu$ would be modified but would not converge to 0 when $p$ was fixed. This motivated us to propose our own relaxation described in Section 5.1 and Appendix B.

# B  IMPLEMENTATION OF EEG AUGMENTATIONS

In this section we review the EEG augmentation that have been proposed and describe their precise implementation, including their case-by-case relaxation mentioned in Appendix A. Their implementation in python is provided in the supplementary material (`braindecode-wip` folder).

The most basic form of data augmentation proposed is the addition of Gaussian white noise to EEG signals (Wang et al., 2018) or to derived features (Yin & Zhang, 2017). This

transformation modifies the waveform shapes in the time domain, while moderately distorting the proportions of different frequencies in the signals spectra. It simply adds the same power to all frequencies equally. It should hence work as an augmentation for tasks where the predictive information is well captured by frequency-bands power ratios, as used for pathology detection in Gemein et al. (2020).

Another type of augmentation is time related transformations, such as `time shifting` and `time masking` (Mohsenvand et al., 2020) (a.k.a. `time cutout` (Cheng et al., 2020)). Both aim to make the predictions more robust, the latter supposing that the label of an example must depend on the overall signal, and the former trying to teach the model that misalignment between human-annotations and events should be tolerated up to some extent.

Similarly, transformations acting purely in the frequency domain have been proposed. Just as `time masking` zeros-out a small portion of the signal in the time domain, narrow bandstop filtering at random spectra positions (Cheng et al., 2020; Mohsenvand et al., 2020) attempts to prevent the model from relying on a single frequency band. Another interesting frequency data augmentation is the `FT-surrogate` transform (Schwabedal et al., 2019). It consists in replacing the phases of Fourier coefficients by random numbers sampled uniformly from $[0, 2\pi)$. The authors of this transformation argue that EEG signals can be approximated by linear stationary processes, which are uniquely characterized by their Fourier amplitudes.

Another widely explored type of EEG data augmentation are spatial transformations, acting on sensors positions. For example, sensors are rotated and shifted in Krell & Kim (2017), to simulate small perturbations on how the sensors cap is placed on the head. Likewise, the brain bilateral symmetry is exploited in Deiss et al. (2018), where the left and right-side signals are switched. `Sensors cutout` (*i.e.,* zeroing-out signals of sensors in a given zone) is studied in Cheng et al. (2020), while Saeed et al. (2021) propose to randomly drop or shuffle channels to increase the model robustness to differences in the experimental setting. Mixing signals from different examples has also been suggested in Mohsenvand et al. (2020).

## B.1 Frequency domain transforms

**Frequency Shift** To shift the frequencies in EEG signals, we carry a time domain modulation of the following form:

$$\texttt{FrequencyShift}(x)(t) := \text{Re}\left(x_a(t) \cdot \exp(2\pi i \Delta f t)\right), \tag{11}$$

with $x_a = x + j\text{H}[x]$ being the analytic signal corresponding to $x$, where H denotes the Hibert transform. At each call, a new shift $\Delta f$ is sampled from a range linearly set by the magnitude $\mu$, where $\mu = 1$ corresponds to the range $[0 - 5\text{Hz})$. This value was chosen after carrying data exploration, based on the observed shifts between subjects in the Physionet dataset.

Given that Equation 11 is completely differentiable on $\Delta f$, we only had to relax the sampling (using the pathwise derivatives trick (Schulman et al., 2015)) to make it differentiable w.r.t $\mu$. More precisely, we define $\Delta f = f_{\max} \cdot u$, where $f_{\max} = 5\mu$ is the frequencies range upper-bound and $u$ is sampled from a uniform distribution over $[0, 1)$.

**FT Surrogate** In this transform, we compute the Fourier transform of the signal $\mathcal{F}(x)$ (using `fft`) and shift its phase using a random number $\Delta\varphi$:

$$\mathcal{F}\left(\texttt{FTSurrogate}(x)\right)[f] := \mathcal{F}(x)[f] \cdot \exp(2\pi i \Delta\varphi).$$

We then transform it back to the time domain. This was reproduced from the authors (Schwabedal et al., 2019) original code.[2] However, in our implementation we added a magnitude parameter setting the range in which random phases are sampled $[0, \varphi_{\max})$, with $\varphi_{\max} = 2\pi$ when $\mu = 1$.

The procedure used to make this transform differentiable is very similar to `frequency shift`,[3] with $\Delta\varphi$ parametrized as $\varphi_{\max}u$ and $\varphi_{\max} = 2\pi\mu$.

---

[2] https://github.com/cliffordlab/sleep-convolutions-tf
[3] It requires using Pytorch version 1.8, which now supports `fft` differentiation.

**Bandstop Filter** This transform was implemented using the FIR notch filter from MNE-Python package (Gramfort et al., 2013) with default parameters. The center of the band filtered out is uniformly sampled between 0 Hz and the Nyquist frequency of the signal. Here, the magnitude $\mu$ is used to set the size of the band, with $\mu = 1$ corresponding to 2 Hz. This value was chosen after some small experiments, where we observed that larger bands degraded systematically the predictive performance on Physionet. This transformation was not relaxed (and hence not available to the gradient-based automatic data augmentation searchers).

## B.2 Temporal domain transforms

**Gaussian Noise** This transform adds white Gaussian noise with a standard deviation set by the magnitude $\mu$. When $\mu = 1$, the corresponding standard deviation was 0.2. Larger standard deviations would degrade systematically the predictive performance in our manual exploration on Physionet. The `Gaussian noise` implementation is straight forward. It's relaxation only included the use of pathwise derivatives for the sampling part: we sample the noise from a unit normal distribution and multiply it by the standard deviation $\sigma = 0.2 * \mu$.

**Time Masking** In this operation, we sample a central masking time $t_{\text{cut}}$ with uniform probability (shared by all channels) and *smoothly* set to zero a window between $t_{\text{cut}} \pm \Delta t/2$, where $\Delta t$ is the masking length. The latter is set by the magnitude $\mu$, where $\mu = 1$ corresponds to $\Delta t = 1s$. This value was chosen because it corresponds roughly to the length of important sleep-related events.

The sampling part of the operation was made differentiable as above. Masking was computed by multiplying the signal by a function valued in $[0, 1]$, built with two opposing *steep* sigmoid functions

$$\sigma^{\pm}(t) = \frac{1}{1 + \exp\left(-\lambda(t - t_{\text{cut}} \pm \frac{\Delta t}{2})\right)},$$

where $\lambda$ was arbitrarily set to 1000.

## B.3 Spatial domain transforms

**Rotations**

In this transform, we use standard sensors positions from a 10-20 montage (using the `mne.channels.make_standard_montage` function from MNE-Python package (Gramfort et al., 2013)). The latter are multiplied by a rotation matrix whose angle is uniformly sampled between $\pm\psi_{\text{max}}$. The value of the maximum angle $\psi_{\text{max}}$ is determined by the magnitude $\mu$, where $\mu = 1$ corresponds to $\frac{\pi}{6}$ radian (value used in Krell & Kim (2017); Cheng et al. (2020)). Signals corresponding to each channel are then interpolated towards the rotated sensors positions. While Krell & Kim (2017); Cheng et al. (2020) used radial basis functions to carry the interpolation, we obtained better results using a spherical interpolation, which also made more sense in our opinion.

The only part that needed to be relaxed to allow automatic differentiation was the rotation angle sampling, which was done as before, with the angle $\Delta\psi$ parametrized as $\psi_{\text{max}}(2u - 1)$, $u$ a uniform random variable in $[0, 1]$ and $\psi_{\text{max}} = \frac{\pi}{6}\mu$.

**Channel Dropout**

Here, each channel is multiplied by a random variable sampled from a relaxed Bernoulli distribution (Maddison et al., 2017) with probability $1 - \mu$, where $\mu$ is the magnitude. This allows to set the channel to 0 with probability $\mu$ and to have a differentiable transform.

**Channel Shuffle**

In this operation, channels to be permuted are selected using relaxed Bernoulli variables as in the `channel dropout`, but with probability $\mu$ instead of $1 - \mu$. The selected channels are then randomly permuted. As the permutation operation necessarily uses the channels

indices, it is not straight-forward to differentiate it, which is why we used a straight-through estimator here to allow gradients to flow from the loss to the magnitude parameter $\mu$.

## C ILLUSTRATION OF THE USEFULNESS OF CLASS-WISE AUGMENTATION ON MNIST

**Motivation** In order to illustrate the potential of CW augmentation, we present in this section a simple example using the MNIST dataset (LeCun et al., 1998). Intuitively, some common image augmentation operations such as horizontal flips or rotations with large angles should not be helpful for handwritten digits recognition: if you rotate a 9 too much, it might be difficult to distinguish it from a 6. However, some digits have more invariances than others. For example, 0's and 8's are relatively invariant to vertical and horizontal flips, as well as to 180 degrees rotations. We used this framework to compare a naive approach of automatic data augmentation search both in a standard and in a CW setting.

**Algorithm** For simplicity, we used a random search algorithm to look for augmentation policies, with operations' parameters discretized over a grid. More precisely, each new candidate subpolicy sampled by the search algorithm is used to train the model from scratch, before evaluating it on the validation set. In terms of search space and search metric, this is equivalent to what is done in AutoAugment (Cubuk et al., 2019), although the reinforcement learning algorithm is replaced by random sampling.

**Search space** Let $N_p$, $N_\mu$ and $N_\mathcal{O}$ denote the number of possible probability, magnitude values and operations in our setting. The search space of both standard and CW policies is potentially huge: $(N_p \times N_\mu \times N_\mathcal{O})^{L \times K \times |\mathcal{Y}|}$, where the number of classes $|\mathcal{Y}|$ is replaced by 1 in the standard case. Given the simplicity of the algorithm used, we tried to reduce the search space as much as possible. Hence, we only considered subpolicies of length $K = 1$ (a single operation) and restrained our problem to the classification of 4 digits only: $4, 6, 8$ and $9$. A pool of four transformations that only depend on a probability parameter (no magnitude) was used in the search: `horizontal` and `vertical flip`, as well as `90` and `180 degrees rotation` (counter-clockwise).

Despite this significant simplification, the search space size for policies made of $L = 5$ subpolicies is still immense: around $10^6$ for standard policies and $1.2 \times 10^{24}$ for CW policies. For this reason, 20,000 trials were sampled in both cases. As an additional baseline, we also trained the same model without any type of data augmentation.

**Experimental setting** After the 4-class reduction, the MNIST training set was randomly split into a training set containing 1000 images per class and a validation set of 3000 images per class. The reduced MNIST test set used to evaluate the final performance has 1000 images per class, as the original one. All transforms were implemented using `torchvision` (Marcel & Rodriguez, 2010). We used the LeNet-1 architecture (LeCun et al., 1998) for the classifier. It was trained using batches of 64 images and Adam optimizer (Kingma & Ba, 2015) with an initial learning rate of $2 \times 10^{-3}$, $\beta_1 = 0.9$, $\beta_2 = 0.999$ and no weight decay. Trainings had a maximum number of 50 epochs and were early-stopped with a patience of 5 epochs. The code that was used to generate this plots are part of the supplementary material (`mnist` folder).

**Results** As seen on Figure C.1, random search converges to the identity policy (no augmentation). This was expected as none of the available operations describe an invariance shared by all four digits. From Figure C.1-right we see that the CW random search does not find a good augmentation for digits 4 and 6 as well. Yet, it is able to recover that digit 8 is invariant to `horizontal flip`. Interestingly, 90 degree rotation of digit 9 is also selected with a small probability. This can be understood, as some people might draw the 9's leg more horizontally than others, so that rotating up to 90 degrees counter-clockwise still preserves in some cases the picture semantics. Figure C.1-left shows that the CW algorithm can find interesting policies in such a constrained setting, but also that it can outperform the baselines. Yet, this happens only after 10,000 trials.

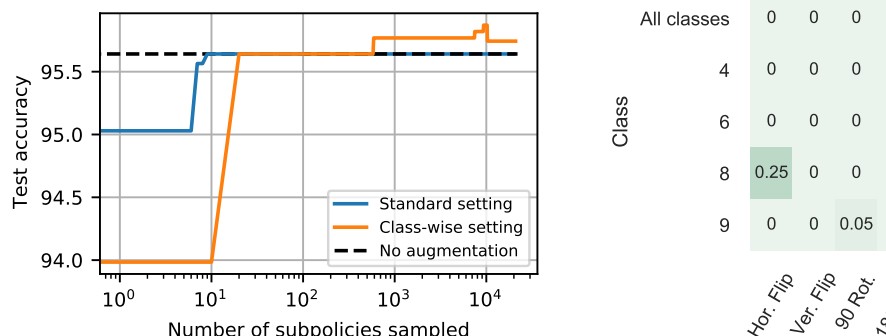

Figure C.1: (**left**) Test accuracy obtained using the subpolicy with higher performance on validation set. (**right**) Probability of applying each operation in the best policy found ($L = 5$). First row corresponds to the standard setting and following rows to the CW setting.

This toy experiment suggests that looking for augmentation strategies depending on the label of the data being transformed can be relevant to improve prediction performance, but also to help discovering some interesting invariances for the task considered. However, despite the extremely simplified setting used, the CW search took 14.4 GPU days (Tesla V100 SXM2). It becomes clear that a more efficient search strategy is required for real-world applications.

# D    EXPERIMENTAL SETTING SHARED ACROSS EEG EXPERIMENTS

In all EEG experiments, learning was carried using the convolutional network proposed in Chambon et al. (2018). The architecture can be found on Table 1 and was chosen as it seemed like a good compromise as a simple yet deep and relevant sleep staging architecture. The first layers (1-4) implements a spatial filter, computing virtual channels through a linear combination of the original input channels. Then, layers 5 to 9 correspond to a standard convolutional feature extractor and last layers implement a simple classifier. More details can be found in Chambon et al. (2018). Other relevant and recent sleep staging architectures are Perslev et al. (2019; 2021); Jia et al. (2021).

| | Layer | # filters | # params | size | stride | Output dim. | Activation |
|---|---|---|---|---|---|---|---|
| 1 | Input | | | | | (C, T) | |
| 2 | Reshape | | | | | (C, T, 1) | |
| 3 | Conv2D | C | C * C | (C, 1) | (1, 1) | (1, T, C) | Linear |
| 4 | Permute | | | | | (C, T, 1) | |
| 5 | Conv2D | 8 | 8 * 64 + 8 | (1, 64) | (1, 1) | (C, T, 8) | Relu |
| 6 | Maxpool2D | | | (1, 16) | (1, 16) | (C, T // 16, 8) | |
| 7 | Conv2D | 8 | 8 * 8 * 64 + 8 | (1, 64) | (1, 1) | (C, T // 16, 8) | Relu |
| 8 | Maxpool2D | | | (1, 16) | (1, 16) | (C, T // 256, 8) | |
| 9 | Flatten | | | | | (C * (T // 256) * 8) | |
| 10 | Dropout (50%) | | | | | (C * (T // 256) * 8) | |
| 11 | Dense | | 5 * (C * T // 256 * 8) | | | 5 | Softmax |

Table 1: Detailed architecture from Chambon et al. (2018), where $C$ is the number of EEG channels and $T$ the time series length.

The optimizer used to train the model above was Adam with a learning rate of $10^{-3}$, $\beta_1 = 0$. and $\beta_2 = 0.999$. At most 300 epochs were used for training. Early stopping was implemented with a patience of 30 epochs. For automatic search experiments, the policy learning rate $\xi$ introduced in (7) was set to $5 \times 10^4$ based on a grid-search carried using the validation set. Concerning the batch size, it was always set to 16, except for CADDA, for which it was doubled to 32, which was necessary to stabilize its noisier gradients. The motivation for such small values was to avoid memory saturation when training Faster AutoAugment

and to preserve the stochasticity of the gradient descent in the learning curve experiments (Figures 3 and G.8), even in very low data regimes.

Balanced accuracy was used as performance metric using the inverse of original class frequencies as balancing weights. The MNE-Python (Gramfort et al., 2013) and Braindecode software (Schirrmeister et al., 2017) were used to preprocess and learn on the EEG data. Training was carried on single Tesla V100 GPUs. The $N_\mathcal{O} = 13$ operations considered were: `time reverse`, `sing flip`, `FT surrogate`, `frequency shift`, `bandstop filtering`, `time masking`, `Gaussian noise`, `channel dropout`, `channel shuffle`, `channel symmetry` and `rotations` around each cartesian axis. In automatic search experiments, `bandstop filter` was not included in the differentiable strategies (Faster AA, DADA, ADDA and CADDA) because we did not implement a differentiable relaxation of it.

## E  Architecture details and comparison between ADDA, DADA and Faster AA

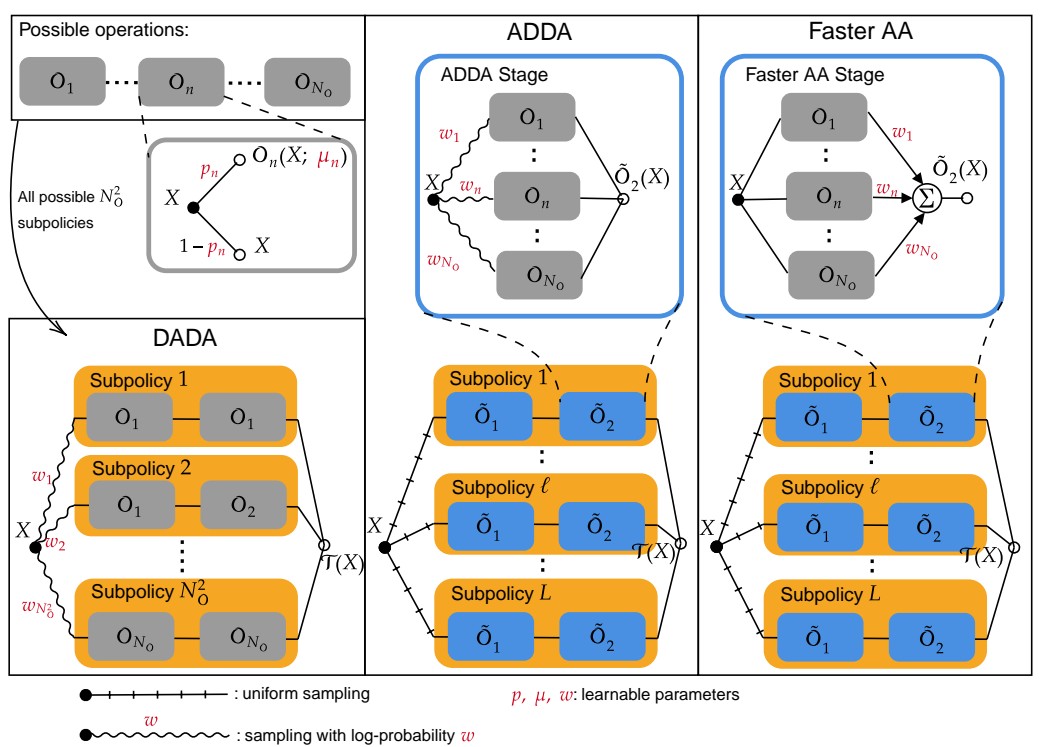

Figure E.2: Different differentiable policy structures of ADDA, DADA and Faster AA. DADA samples whole subpolicies according to learnable log probabilities, whereas ADDA and Faster AA sample from $L$ subpolicies with uniform probability. Also, while subpolicy are sequences of parametrized operations in DADA, their are made of stages in ADDA and Faster AA. In Faster AA stages, a convex combination of all possible operations is computed. In ADDA, however, operations are sampled with learnable log probabilities.

As depicted on Figure E.2, DADA policies will sample a subpolicy according to a Gumbel-softmax distribution parametrized by learnable weights $w$:

$$\mathcal{T}_{\text{DADA}}(X) = \tau_i(X), \qquad \text{with } i \sim \text{Gumbel-Softmax}(\{1, \dots, (N_\mathcal{O})^K\}; w).$$

Each subpolicy $\tau_i$ is just a sequence of operations differentiable wrt its parameters (*cf.* Appendix A). However, in ADDA and Faster AA, only a predetermined number of $L$ subpolicies

are considered and sampled uniformly:

$$\mathcal{T}_{\mathrm{ADDA}}(X) = \tilde{\tau}_i, \qquad \text{with } i \sim \mathcal{U}(\{1, \ldots, L\}).$$

In this case, subpolicies $\tilde{\tau}_i$ are sequences of stages $\tilde{\mathcal{O}}_k$, as described in equation (6). While these stages compute a convex combination of all possible operations in Faster AA, using learnable weights $w$ mapped through a softmax function $\sigma_\eta$, ADDA stages carry a differentiable sample (i.e. $\sigma_\eta$ is exactly equal to 1 for exactly one out of $N_\mathcal{O}$ in (6)). In this case, the weights correspond to the log probabilities of sampling each operation.

## F  COMPLEMENTARY AUTOMATIC DATA AUGMENTATION RESULTS ON EEG SLEEP STAGING

**Class-wise policies selected by CADDA and other approaches**

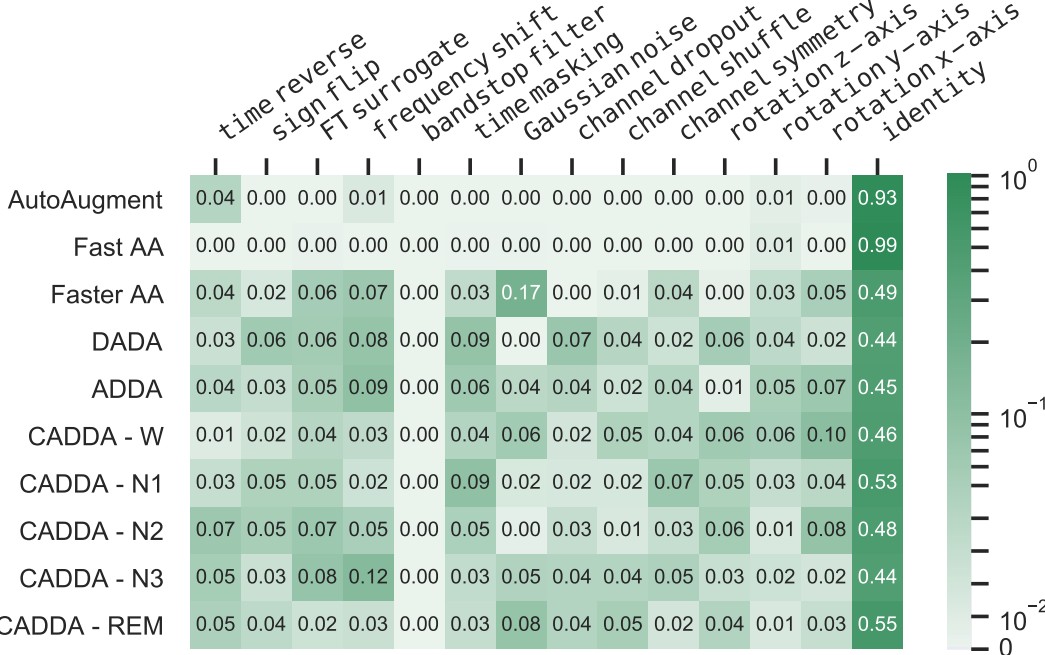

Figure F.3: Probability of applying each operation in the best policies obtained by the different search strategies. For each method, probabilities of all subpolicies are grouped. Methods based on differentiable augmentation policies discover relevant augmentation strategies such as `time reverse`, `sign flip`, `FT surrogate` etc. CADDA suggests that each class benefits differently from the different strategies.

**Complete comparison of gradient-based and gradient-free methods in the class-agnostic setting**

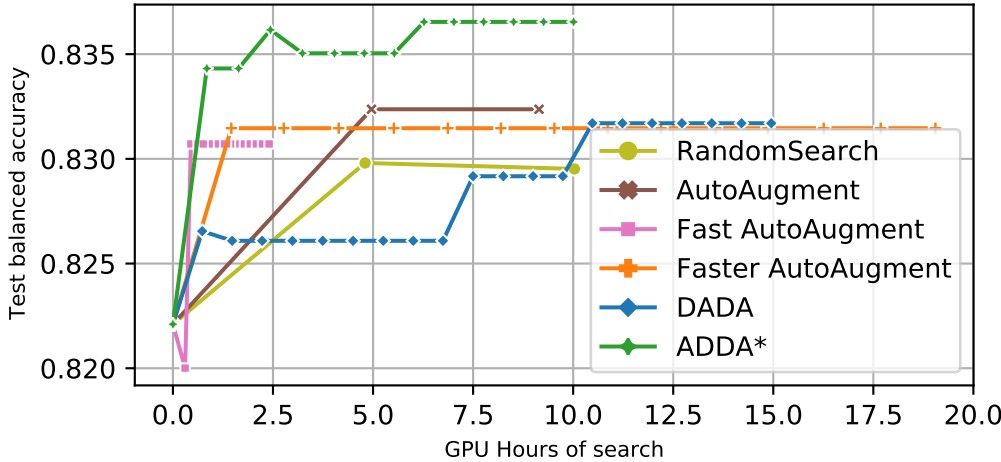

Figure F.4: Median performance obtained by different automatic data augmentation strategies as a function of the computation time. ADDA outperforms the state-of-the-art in speed and final accuracy in the class agnostic setting.

**75% confidence interval**

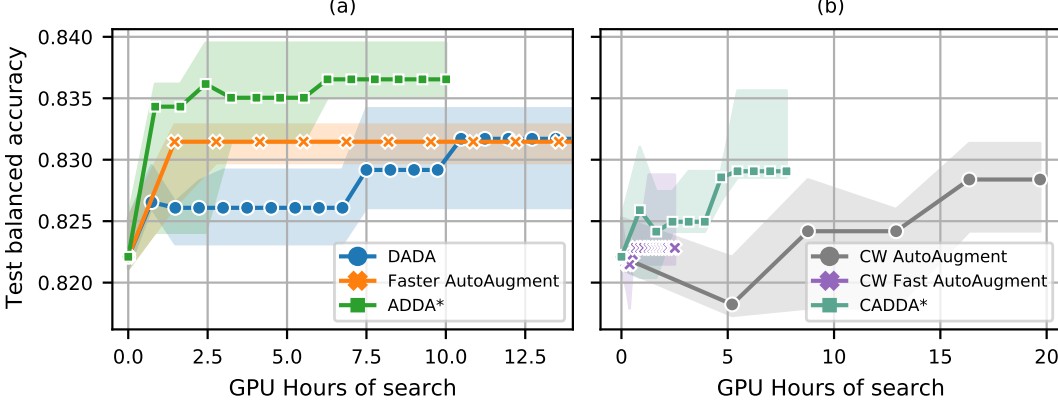

Figure F.5: Median balanced accuracy (over 5 folds) on the test set of different ADA strategies as a function of the computation time. This figure is the same as Figure 4 with error bars representing 75% confidence intervals estimated with bootstrap.

**Macro F1-score plots**

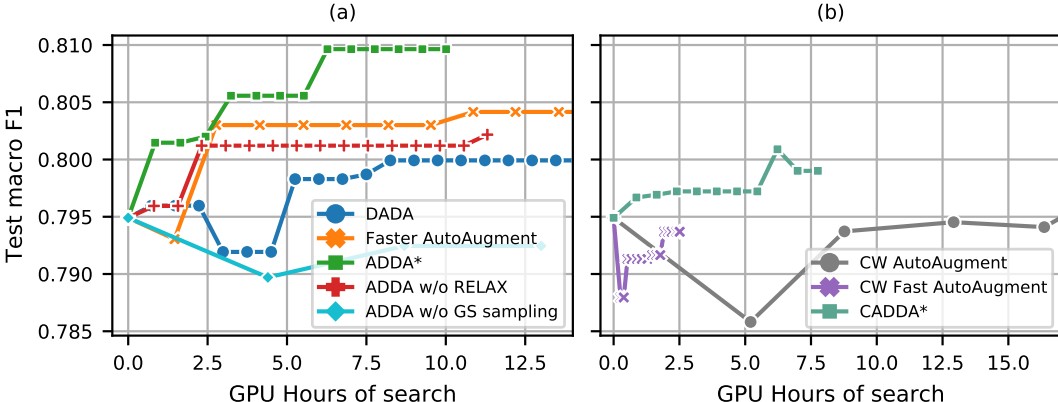

Figure F.6: Median macro F1-score (over 5 folds) on the test set of different ADA strategies as a function of the computation time. (a) Class-agnostic setting: ADDA is 40% faster than Faster AA and leads to a final performance 0.5% higher. It also outperforms DADA by 1%. (b) CW setting: CADDA is 5x faster than AutoAugment and achieves higher performance than gradient-free methods.

**Impact of data regime on density matching approaches**

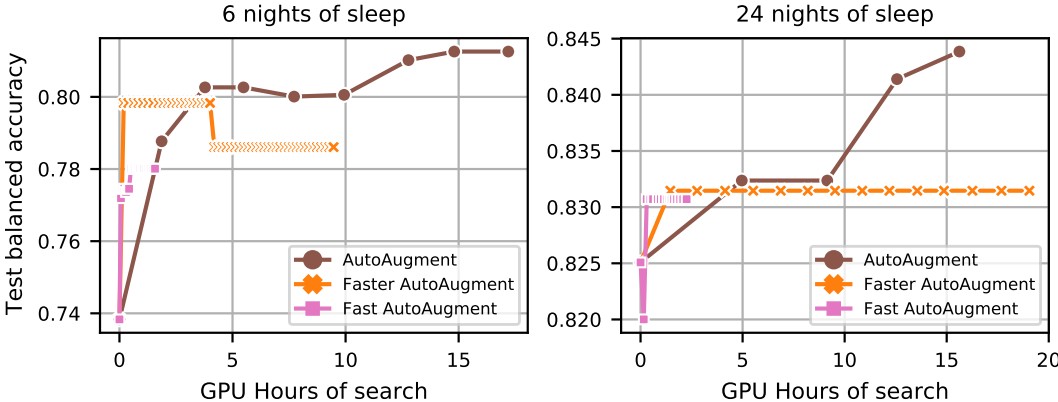

Figure F.7: Median performance obtained by different automatic data augmentation strategies as a function of the computation time with 6 and 24 nights of sleep for training and validation. With 6 nights, Fast AutoAugment fails to learn relevant augmentation policies and is not competitive against AutoAugment and Faster AutoAugment. This effect is mitigated with 24 nights, when the classifier is trained on sufficient data to be able to capture relevant invariances.

## G    Further manual exploration details

### G.1    Manual search results on MASS dataset

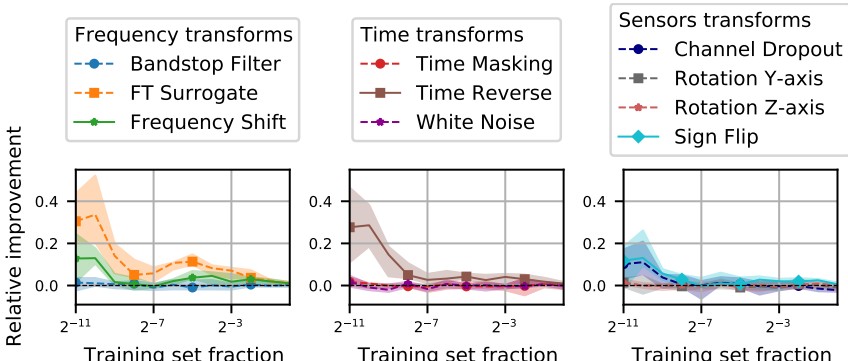

Figure G.8: Median performance gains obtained by individual augmentation strategies on the MASS dataset. Best transforms obtained on Physionet (*cf.* Fig. 3) give here consistent improvements, while sensors transformation like Channel Dropout are more relevant when using 6 channels. Here also using more training data mitigates the need for data augmentation, although it still improves the predictive performance.

### G.2    Magnitudes fine-tuning for manual exploration

In this section we explain the fine-tuning of magnitudes used for the manual exploration results presented in Section 6.

A first training with probability $p = 0.5$ and magnitude $\mu = 0.5$ was carried over the same training subsets as in Figure 3 and Figure G.8. Then we selected the subset where the effects of augmentation were maximized ($2^{-11}$ times the original training set for both Physionet and MASS datasets). Finally, for each operation, we carried a grid-search over the magnitude parameter (with fixed probability $p = 0.5$) with the selected training subsets (over 5-folds, as described in Section 6). The result of the grid-search can be seen on Figure G.9 and Figure G.10.

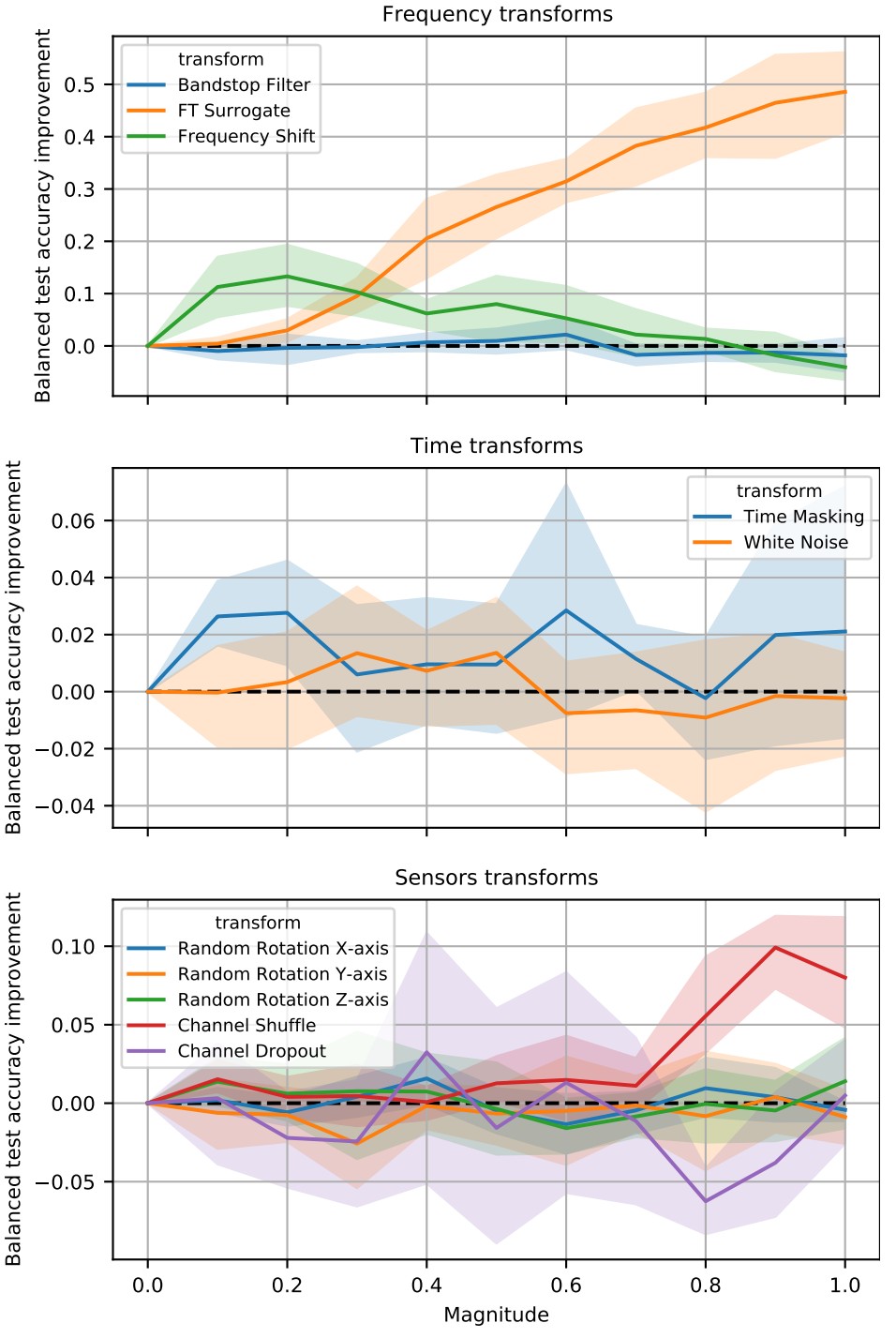

Figure G.9: Magnitudes grid-search results for Physionet dataset. Balanced accuracy values are relative to a training without any augmentation.

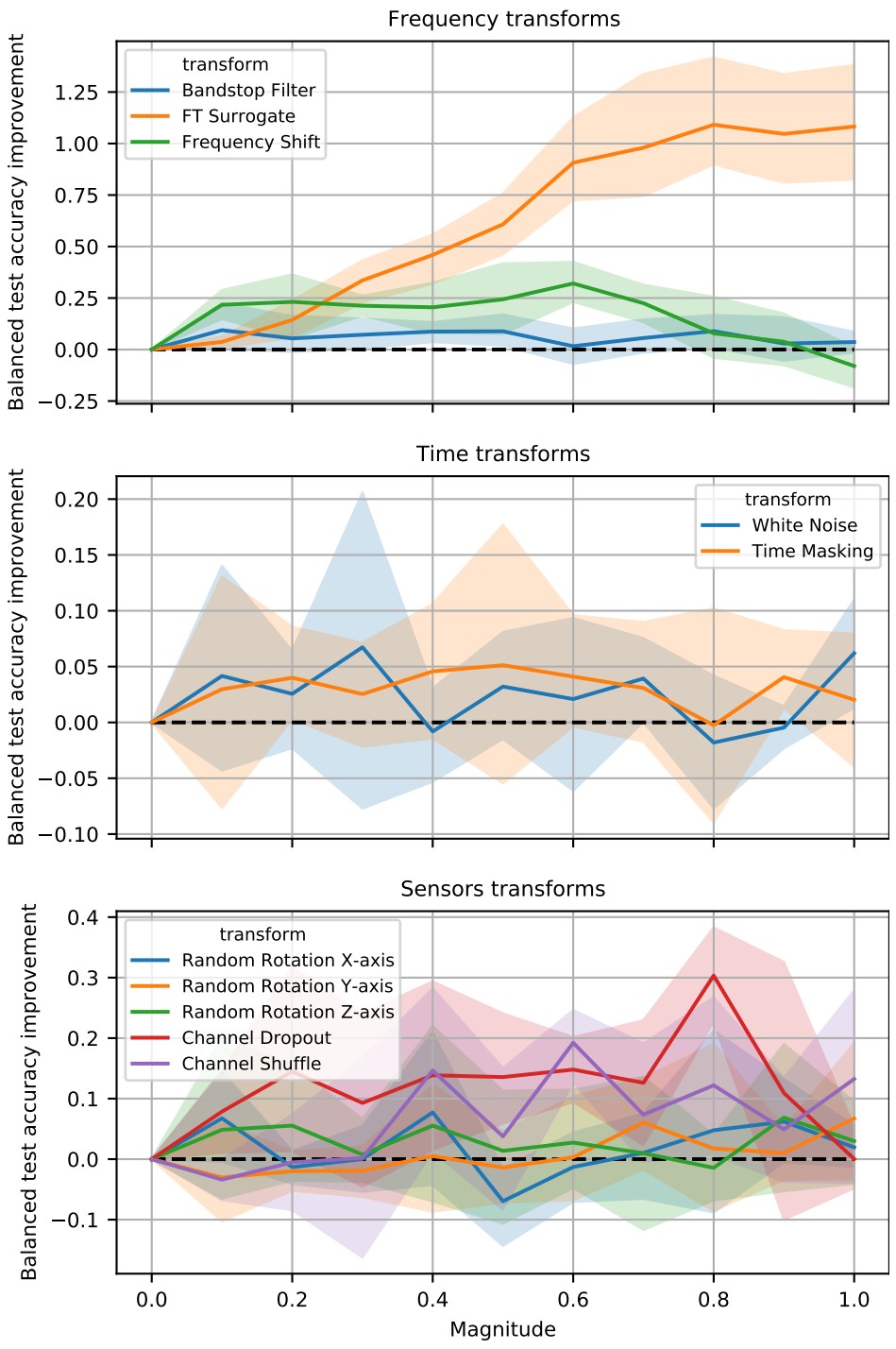

Figure G.10: Magnitudes grid-search results for MASS dataset. Balanced accuracy values are relative to a training without any augmentation.

