# OpenReview forum: "CADDA: Class-wise Automatic Differentiable Data Augmentation for EEG Signals"
_ICLR.cc/2022/Conference — ICLR 2022 Poster_

### Official Review · Reviewer_NPRR · 2021-10-30

**Correctness:** 3
**Technical Novelty And Significance:** 2
**Empirical Novelty And Significance:** 2
**Recommendation:** 5
**Confidence:** 3

**Main Review:**

Pros:

+ This is an interesting work. The examples of object recognition and sleep staging are very intuitive and demonstrates the motivation of exploring the CW data augmentation approaches.

+ The authors designed some data enhancement methods for EEG. The new EEG data augmentation **frequency shift** introduced in Section 3 makes sense, as Figure 1 illustrates the fact that applying a 0.5 Hz frequency shift transform to one subject can lead a power spectrum density more similar to another.
+ The paper shows adequate experiments and provides detailed supplementary material.

Cons:

+ It is mentioned that the three new EEG data augmentations, namely time reverse, sign flip and frequency shift, are operations acting on the time, space and frequency, respectively. But I'm confused about why sign flip can be categorized as a spatial operation. Please clarify it.

+ Since data augmentation is usually employed to solve the class imbalance problem, which is a great challenge for sleep stage classification, macro- and micro-F1 score should also be presented in the performance comparison experiments.

+ The performance of ADDA and CADDA are compared, which shows that CADDA does not improve over ADDA. But why the comparison of class-wise(CW) and class-agnostic(CA) settings of other baselines is not conducted? I wonder whether the CW setting of other methods can contribute to a better performance than the CA setting.

+ It is mentioned that "For each run, when the final retraining validation accuracy improves compared to previous retraining, we report the new test accuracy obtained by the retrained model (otherwise, we keep the previous value)", but the 5th point of ADDA in Figure 4(a) is lower than the 4th point, which seems to be contradictive. Please clarify it.

+ More recent methods, especially the ones work on time-series data augmentation (e.g. [1]), should be compared with the proposed method.

+ Although the author focuses on solving problems in the field of data enhancement. But sleep staging is an important application of the author. I hope that the method proposed by the author can be compared with some of the latest sleep staging methods, such as [2] and [3].

  [1] Fons E, Dawson P, Zeng X, et al. Adaptive Weighting Scheme for Automatic Time-Series Data Augmentation[J]. arXiv preprint arXiv:2102.08310, 2021.

  [2] Perslev M, Jensen M H, Darkner S, et al. U-time: A fully convolutional network for time series segmentation applied to sleep staging[J]. arXiv preprint arXiv:1910.11162, NeurIPS, 2019.

  [3] Jia Z, Lin Y, Wang J, et al. SalientSleepNet: Multimodal Salient Wave Detection Network for Sleep Staging[J]. IJCAI, 2021.




**Summary Of The Paper:**

The work explores the problem of automatic data augmentation beyond images which is still rarely considered in the literature. The authors provided novel transforms for EEG data, and a state-of-the-art search algorithm.

**Summary Of The Review:**

They propose a new differentiable relaxation of the problem. Some details of the evaluation and comparison of experiments need to be explained or clarified.

---

> ### Author Response · Authors · 2021-11-15
> **Answer to reviewer NPRR**
>
> We thank the reviewer for judging that our paper is “novel”, “interesting” and to have “adequate experiments”. We shall address each concern expressed by the reviewer in the same order below:
>
> 1. “How come sign flip is a spatial transformation”
>
> We thank the reviewer for this constructive question. Indeed, sign flip is a bit borderline within the category of spatial operations, because the spatial positions of the electrodes are not actually used. The general intuition behind this augmentation is that electric charges are equally likely to be moving towards deep or superficial layers of the cortex. One of the reasons we chose to put this operation in this category is that it can also be interpreted as switching the main electrodes and the reference electrodes in some datasets such as SleepPhysionet: Cz-FPz instead of FPz-Cz and Oz-Pz instead of Pz-Oz. We have added this remark in the paper (section 3, page 4).
>
> 2. “Also plot F1 scores in experimental results”
>
> Indeed, data imbalance is a great challenge in sleep staging and both MASS and SleepPhysionet suffer from it. For this reason, all results are plotted in terms of balanced accuracy. Furthermore, we have added in the appendix a second version of the same plots from figure 4 in terms of macro F1 score (figure G.6). This new metric does not change the conclusions of the paper.
>
> 3. “Why not plot other class-wise algorithms on figure 4b”
>
> We chose to plot on figure 4b class-wise Fast AutoAugment and AutoAugment because they are the best gradient-free methods in a class-agnostic setting in terms of speed and accuracy respectively (see figure G.4). Likewise, we have plotted class-wise ADDA because it beats all other gradient-based methods on figure 4a. We did not want to overcrowd the plot, whose main message is to show that gradient-based automatic data augmentation is necessary in the class-wise setting. Moreover, we had to make choices given the high compute time of all these methods and we chose to ignore the less performing methods based on figure 4a.
>
> 4. “Why some metrics decrease”
>
> This is a good remark and we understand the confusion. Points in the plot are allowed to decrease because they represent the accuracy on the test set, while we use the validation set to decide whether to keep or not the new performance. Hence, while the validation performance can only increase, the test performance can potentially decrease.
>
> 5. “Additional comparison to [1]”
>
> It is difficult for us to add a comparison to this new method as the latter has no open-source code to be used. Comparing to [1] would hence require a complete reimplementation from scratch.
> Furthermore, we would like to add that we are already comparing to a large number of relevant SOTA methods:
> * 5 automatic data augmentation methods using 13 augmentation operations each,
> * 2 of these are very recent (2021) and very close to our method (only gradient-based methods we are aware of).
>
> We agree that this reference is relevant to our work. It is now cited in the updated version of the manuscript.
>
> 6. “Comparisons with different architectures”
>
> We thank the reviewer for this remark. In order to have a clear analysis of the existing automatic data augmentation methods in the new EEG setting, we found that we should use a relevant yet relatively simple neural network architecture, which is why we used Chambon et. al. 2018 throughout all experiments. Given the already large number of experiments presented in the paper (many with results postponed to the appendix), we would like to argue that depicting results with this standard architecture is fair, especially since the core comparisons in the paper are between alternative automatic augmentation methods. We will certainly consider this question in a future journal version of this work. We have added more references to recent sleep staging architectures in our paper, including [2] and [3].

---

### Official Review · Reviewer_18fi · 2021-10-31

**Correctness:** 3
**Technical Novelty And Significance:** 3
**Empirical Novelty And Significance:** 3
**Recommendation:** 8
**Confidence:** 4

**Main Review:**

- This paper investigates gradient-based automatic data augmentation algorithms amenable to class-wise policies with exponentially larger search spaces.

- The idea of the proposed method, CADDA, is interesting. Indeed, the paper shows that the exploration of gradient-based search approaches allows to select policies more efficiently in a huge search space.

- The novelty of the proposed method is on the novel operations relaxation and the estimation of the augmentation policy gradient.


**Summary Of The Paper:**

This paper studies gradient-based automatic data augmentation algorithms amenable to class-wise policies with exponentially larger search spaces. It presents a method, called  CADDA, to address the problem of automatic data augmentation with application on EEG signals. The proposed method achieves good results for EEG sleep stage classification.

**Summary Of The Review:**

The paper is clear and well presented. The idea of the proposed method seems interesting. The experimental results show the effectiveness of the proposed method. A supplementary material was given to show more detail and results.

---

> ### Author Response · Authors · 2021-11-15
> **Answer to reviewer 18fi**
>
> We thank the reviewer for their positive feedback and for judging that our paper is “clear”, “well presented” and “interesting”.

---

### Official Review · Reviewer_EdsK · 2021-11-02

**Correctness:** 3
**Technical Novelty And Significance:** 2
**Empirical Novelty And Significance:** 3
**Recommendation:** 6
**Confidence:** 3

**Main Review:**

Strengths:
1. The work done by authors in the field of data augmentation for EEG data is novel and interesting. The improvement achieved by these augmentations is significant and could be important for building robust models. Especially, the frequency shift augmentation can be important for building models that can overcome the domain-shift problem due to inter-subject variability. The proposed augmentations are also given high weights by the automatic algorithms (Fig. F.2) which are interesting.

2. Authors' preliminary experiments on investigating the usefulness of class-wise augmentations are also interesting. As mentioned by the authors, the field is relatively less explored (even more so for EEG data). Hence, the first steps taken in this direction are significant.

Areas of improvement:
1. Authors shows experimental results only for two datasets which are not standard. To alleviate our concerns about the quality of computational experiments and to strengthen authors' claims, it would be great if the authors can provide results on standard datasets (Imagenet, SVHN, CIFAR-100, or CIFAR-10) and show an improvement over previous methods.

2. Model architecture details are missing. It would be great if the authors can include these details. Additionally, previous works like DADA have shown that their method performed well irrespective of model architecture or downstream tasks. It would be great if authors can include additional experiments to show that their method also performs well for different model architectures and downstream tasks (classification, segmentation, object detection, etc.)

3. As per the authors, the core difference between their method and DADA is that DADA only samples the whole sub-policies whereas they sample operations within each sub-policy. But, according to my quick overview of DADA, in addition to sampling whole sub-policy, they do sample operations within each sub-policy (see sections 3.2 and 3.4 of [DADA](https://arxiv.org/pdf/2003.03780.pdf)). It would be helpful for the readers if authors can concretely differentiate their method from DADA using mathematical equations for both methods. They can include it in the appendix. I think this would help to understand the novelties of (C)ADDA over DADA.

4. Some other algorithms like PBA and OHL-AA are missing from the comparisons. I would encourage authors to include them to strengthen their claims. Additionally, instead of keeping Randomsearch in the appendix plot (Fig F.3), I would include it in the plot in the main text (Fig. 4(a)) so that readers can get some baseline reference. A related question, why was Fast AutoAugment (pink in Fig. F.3) not shown in the main plot? It has better performance in terms of computation time and accuracy than DADA.

5. Since the difference in the performance of different methods is small (~1%), I would like to see 95% confidence intervals for the performance measure (by bootstrapping the test set).

6. For CW methods, just like CW-FastAuto Augment and CW-Auto Augment, what happens if we implement CW-DADA?

7. Fig 2. b shows improvements only for two augmentations, what happens for other augmentations?

8. As mentioned in the strengths, although the idea of class-wise augmentation is interesting, CADDA does not outperform ADDA and many other methods. So should this paper be titled based on CADDA? Wouldn't it be misleading for the readers as they would expect to see a Class-wise algorithm that outperforms all other methods?

**Summary Of The Paper:**

The paper proposes an automatic differentiable data augmentation algorithm for EEG data that outperforms existing methods. They also propose novel augmentations for EEG that help the model to train better in low-labeled data regimes. They also show preliminary results showcasing that class-wise augmentation can be better than class agnostic augmentations for EEG data.

**Summary Of The Review:**

Although the paper has some great (but minor) contributions in terms of the novel augmentation for EEG data, there are still many aspects in which the paper can be improved. As per the current state of the paper, my recommendation would be a score of 5 (marginally below the acceptance threshold). If the authors address the above-mentioned concerns and strengthen their claims with additional experiments, I will be happy to update my score.

---

> ### Author Response · Authors · 2021-11-15
> **Answer to reviewer EdsK**
>
> We thank the reviewer for judging that our paper is “significant”, “novel” and “interesting”, as well as for the very constructive criticism. We shall address each concern expressed by the reviewer in the same order below:
>
> 1. “Why not add experiments on computer vision datasets”
>
> Indeed, we understand that previous works focus on standard computer vision datasets such as Imagenet, SVHN, and CIFAR and we agree that benchmarking our method on those would be interesting too. However, we would like to highlight that one of the main points of our paper is also to test these existing methods on other datasets from other application fields. Indeed, automatic data augmentation beyond computer vision is under-explored while it can have huge impact, as augmentations that work are highly unknown with other types of data, such as EEG signals. Moreover, existing implementations could not be straightforwardly used in our context as they are oriented towards augmenting image datasets and we had to re-implement them to be able to do our benchmarks. Adapting implementations of existing data augmentation algorithms and demonstrating their usefulness in this context is a contribution on its own. Also, please note that both SleepPhysionet and MASS are very well-known standard datasets in the EEG community. We plan on extending our experiments to SVHN and CIFAR, as well as releasing a standard implementation of all these techniques in a follow-up journal paper.
>
> 2. “Missing model architecture details and additional experiments”
>
> We agree with the reviewer that the sleep staging model architecture was not described in the paper. We have added in the appendix section D a detailed description of the neural network architecture used, with a footnote in the core text referring to it on page 5. Please let us know if this addresses your concern correctly.
>
> Concerning further experiments with other architectures, we agree that it would strengthen our results. We chose this architecture because it is relevant and deep yet relatively simple. Given the already large number of experiments presented in the paper (many with results postponed to the appendix), we would like to argue  that depicting results with this standard architecture is fair, especially since  the core comparisons in the paper are between alternative automatic augmentation methods. We will certainly consider this question in a future journal version of this work.
>
> 3. “Clarify differences between DADA and ADDA policy architectures”
>
> We thank the reviewer for this very constructive remark. What is called “operations sampling” in DADA is just the samping of a relaxed Bernoulli random variable used to decide whether to apply an augmentation operation or not within a subpolicy (see figure 2, page 5 of DADA). This sampling is common between DADA, Faster AutoAugment and our method, ADDA. The difference between ADDA and DADA lies on the other sampling: DADA samples whole subpolicies (sequences of operations), while ADDA (as Faster AutoAugment) has a fixed number of subpolicies where at each stage a different operation is sampled. The latter occurs before the sampling of the relaxed Bernoulli.
>
> We agree that concrete equations and a figure showing more explicitly the architectural differences between DADA and ADDA would improve the paper clarity. Both have hence been added to the appendix of the revised paper (section F) and are referenced at the end of subsection 5.1. Please let us know whether it addresses your concern.
>
> 4. “Why not add PBA, OHL, RandomSearch and Fast AA to the main figure 4 a”
>
> While we agree that comparing to PBA and OHL could strengthen our empirical results, we believe that we are already comparing to a fair number of methods (the most relevant ones in our opinion). Furthermore, adding these two new methods to the benchmark would imply adapting their available code to EEG datasets, which is non trivial from a software viewpoint as they are highly tailored for computer vision datasets (cf. answer 1).
>
> Furthermore, figure 4a has two main messages, which are:
> - ADDA can outperform previous gradient-based methods,
> - ADDA works better because it mixes the best ideas from DADA and Faster AutoAugment (ablation study).
>
> To clearly convey these messages, we needed to plot 5 curves and thought that adding all SOTA methods tested would overcrowd the figure and blur the messages. That is why Randomsearch and Fast AutoAugment were deferred to the appendix and we already refer to the more complete figure G.4 in the main text. However, if the reviewer thinks the paper would benefit from moving these curves to the main plot, we can do it.

---

> > ### Author Response · Authors · 2021-11-15
> > **Rest of the answer to reviewer EdsK**
> >
> > 5. “Error bars”
> >
> > The reason figure 4 does not depict error bars is mainly because it becomes less readable. We have hence added a new version of figure 4 with error bars representing the 75% confidence interval (bootstrapped from 5 folds) to the appendix (figure G.5). A 95% confidence interval cannot be properly estimated with 5 folds. The ablation study was removed to emphasize the improvement of our contribution (ADDA) over the SOTA (Faster AA and DADA) and improve readability of the figure. Please, let us know if this addresses your concern correctly.
> >
> > 6. “Why not add class-wise DADA to figure 4b”
> >
> > The message of figure 4b is to show that gradient-based automatic data augmentation is necessary in the class-wise setting due to the high dimensional search space. Hence, we chose to plot class-wise Fast AutoAugment and AutoAugment because they are the best gradient-free methods in a class-agnostic setting for speed and accuracy (see figure F.3). Likewise, we have plotted class-wise ADDA because it beats DADA and the others on figure 4a. Not only did we not want to overcrowd the plot, but also we had to make choices given the high compute time of all these methods.
> >
> > 7. “Why not compare the class-wise policy found manually in figure 2b to other class-agnostic augmentations?”
> >
> > Thanks for your remark. The idea of figure 2b is to show that by mixing two augmentation operations (sign flip and Gaussian noise) into a class-wise augmentation, we can obtain better results than using those same augmentations in a class-agnostic setting. This offers empirical evidence motivating the following contributions of the paper, which is to develop an algorithmic solution for learning such class-wise policies. Note that the suggested comparison would lead to a combinatorial explosion of possible policies to evaluate, which is precisely what the paper is trying to avoid.
> >
> > 8.“CADDA in the paper title”
> >
> > Indeed, we are aware that, in spite of what the paper title suggests, the empirical results of CADDA are not the main take-home messages for the reader. The paper contributes novel augmentation operations for EEG, a class-wise formalism for data augmentation and an efficient gradient-based search algorithm evaluated on an original application beyond computer vision. If approved by the reviewer, we hence propose to change the title to “Automatic Differentiable Data Augmentation for EEG Signals in a Class-wise Setting”.

---

> > ### Comment · Reviewer_EdsK · 2021-11-29
> > **Thank you for addressing my questions/concerns**
> >
> > Thank you for addressing my questions. Given that the authors have addressed most of my questions (although not all), I have decided to upgrade my rating from 5 to 6.
> >
> > Regarding comment 4 (Adding PBA and OHL), I would recommend the authors to add it to the paper (if not in the main text, at least in the appendix). This would strengthen the paper.

---

### Official Review · Reviewer_jp1B · 2021-11-06

**Correctness:** 4
**Technical Novelty And Significance:** 2
**Empirical Novelty And Significance:** 2
**Recommendation:** 6
**Confidence:** 2

**Main Review:**

Strengths:

1) The paper provides a strong foundation of related work and solid experiments to validate the improvement from their idea.
2) The way of using AutoAugment in EEG data is interesting and novel.

Weakness:

1) This paper is very hard to follow. The writing will require significant reorganization. It seems Section 3-5 are all of their proposed methods, but it also contains a significant amount of content of former work. I think it would be really great to have a method/approach section to talk about the proposed method. Also, section 4 also includes some evaluation details, which makes me very confused to understand. Please cluster all the evaluation details into Section 6. I think the main contribution of this paper should be from section 5, please try to highlight it.

2) There is only a little accuracy gain in this paper (1.4%). Could you please provide more justification for this?


**Summary Of The Paper:**

THis paper proposes a special version of AutoAugment to search class-wise data augmentation policies for EEG data. The main contribution of this paper is a novel differentiable relaxation algorithm on EEG data (ADDA) that significantly efficiency of policy search. Through the EEG sleep staging task, the paper shows they achieved SOTA with 40% speed on efficiency and 1.4% accuracy increase.


**Summary Of The Review:**

This is an interesting paper in machine learning and healthcare. The approach introduces novelty in ML and has a potential to broader healthcare applications. However, the paper requires more work to improve writing quality.

---

> ### Author Response · Authors · 2021-11-15
> **Answer to reviewer jp1B**
>
> We thank the reviewer for judging that our paper is “strong”, “novel” and “interesting”. The main criticisms denoted by the reviewer is about the structure of the paper (1) and the moderate accuracy gain (2).
>
> (2) Concerning the second point, as better viewed on figure G.4, the proposed method (ADDA) is capable of achieving the same level of accuracy as the top performer (AutoAugment) 10 hours earlier, i.e. 10 times faster. We hence believe that our method represents some true improvement over the state-of-the-art since other existing methods are either slower or less accurate than ours.
>
> (1) Concerning the first point, the structure of the paper was chosen to better present our contributions with respect to the state-of-the-art. As our contributions span over three different domains, we needed to clarify related work in each of these domains before presenting each contribution:
> * In section 3 we present the novel augmentation operations that we propose for EEG signals.
> * In section 4, we present our class-wise extension to the existing augmentation framework proposed in AutoAugment, and have hence to recall this framework.
> * In section 5, we present our new automatic data augmentation algorithm (ADDA) and need to introduce in details the elements borrowed from previous works, e.g. part of the policy structure from Faster AutoAugment and the training algorithm used in DADA.
>
> For this reason, it seemed difficult to group all our contributions in a single section. In an attempt to better clarify in the core text what is original work and what is previous work, we have separated each of these elements in paragraphs with clear titles such as: “Background on auto augmentation framework” (page 5), “Novel class-wise subpolicies and policies” (page 5), “Background on probabilities and magnitudes relaxation” (page 6) and “Novel operations relaxation” (page 6). Please, let us know if this addresses the first part of your concern.
>
> Concerning the experiment in section 4, we chose to detach it from the experiments section 6 because it is a toy experiment which motivates class-wise augmentation and shows that we cannot tune them manually. It hence illustrates the need for automatic search algorithms which are studied in the next sections 5 and 6. That being said, we can move it to section 6 and just add a reference to it at the end of section 4 if the reviewer thinks it is clearer this way.

---

### Author Response · Authors · 2021-11-15
**General answer**

We would like to thank all the reviewers for the time spent analyzing and improving our work with constructive criticism. Individual answers can be found below each of your reviews. Please note that a revised version of the manuscript has been uploaded as well. Modifications wrt the initial submission were colored in green.

---

### Author Response · Authors · 2021-11-26
**Feedback**

Dear reviewers,

As we are reaching the end of the reviewing process, we would greatly appreciate getting some feedback on our answers to your concerns and our revised manuscript submitted now two weeks ago. This would allow us to clarify any questions that you might still have, which would ensure a fair and complete discussion about our work.

The authors,

---

### Decision · Program_Chairs · 2022-01-20

**Decision:**

Accept (Poster)

**Comment:**

This paper is close to the borderline, but I think it is good enough that I recommend its acceptance. Although there were some problems raised by the reviewers, the authors managed to successfully address a majority of them. Having said that, I still recommend that the authors carefully analyze the reviews again and make sure that they incorporated reviewers' comments in the final version of the paper. A lot of them were constructive and might improve the quality of the paper.